# Moisture Sorption of Wood Surfaces Modified by One-Sided Carbonization as an Alternative to Traditional Façade Coatings

**Maija Kymäläinen** [1,*], **Jakub Dömény** [2] **and Lauri Rautkari** [1]

1   Department of Bioproducts and Biosystems, School of Chemical Technology, Aalto University, P.O. Box 16300, FI-00076 Aalto, Finland
2   Department of Wood Science and Technology, Faculty of Forestry and Wood Technology, Mendel University in Brno, Zemědělská 3, 61300 Brno, Czech Republic
*   Correspondence: maija.kymalainen@aalto.fi

**Abstract:** Surface carbonization, or charring, of wooden cladding boards is a promising, low impact process that can substitute inorganic coatings. The char surface is inert and hydrophobic and possibly a long-lasting solution for exterior uses. To determine the performance of surface-charred wood, several sorption experiments were established. Sapwood of two hardwoods (aspen, birch) and two softwoods (pine, spruce) were used as the experimental material, modified by contact charring and gas flame charring, including surfaces coated with oil for further protection. The results showed that flame charring modification is equal to a double layer of acrylic paint on primer in terms of permeability on all wood species, with higher moisture exclusion efficiency at high relative humidity. Contact charring modification presented much improved properties in comparison to both flame charring and acrylic paint in all implemented sorption experiments, but the dimensional stability was poor with strong cupping in wetting. However, hardwoods and especially birch exhibited less dimensional distortion than the other investigated species, and oiling further reduced the cupping. The contact charring modification produced more repeatable results with less impact from substrate and all wood species benefitted, whereas the flame charring modification is more dependent on inherent properties of the wood species, and does not seem to suit aspen as well as birch, and pine as well as spruce, although oiling affected the observed results.

**Keywords:** claddings; permeability; sorption; surface modification; wood; wood char

## 1. Introduction

Surface-charred wood products have recently been brought successfully on the markets aided by a unique, natural look favored by consumers [1]. Despite being based on an ancient technique known as *yaki sugi*, knowledge on durability and service life of this type of cladding material is still rather scarce. To tackle this gap, studies have been published on traditional manufacturing methods [2,3], gas flame-charred surfaces [4,5], and on several variants of experimental contact charring methods [6–11]. As charring is implemented only on the surface, it is justified to compare this modification to a coating. The purpose of a coating is to act as a barrier and protect the wood from adverse effects of weathering, as well as control the moisture movement within. One of the most important aspects is the elimination of excess moisture to minimize dimensional alterations such as cupping and warping that cause cracking of wood and loosening of attachments. As a naturally hygroscopic material, wood interacts with the relative humidity (RH) of the environment, and a moisture flux into the coating and the wood is formed. The flux is caused by adsorption of water molecules into molecular sites and/or diffusion of water molecules through free volumes or along particle interfaces [12,13]. Liquid water, on the other hand, moves through the wood by capillary pressure. A coating is expected to block the wood capillaries and therefore significantly hinder capillary flow [13], but in char the pores and capillaries

exist as they would in unmodified, uncoated wood. Thus, the protective functions must come from the modification itself. Uncoated wood may also be used in claddings, but careful design must be implemented to avoid uneven weathering and structural damage. The advantages of uncoated claddings are, e.g., the greatly reduced need for maintenance, natural look, and reduced environmental loads from upkeep and re-coating. An uncoated façade will (if properly installed) allow the unhindered ingress and egress of water vapor that allows the wetted structure to dry out. This permeability of water vapor is a crucial property also affecting the performance of wood finishes [14]. However, vapor permeability (the rate of flow of water vapor through the substrate and through the substrate plus coating) may in some coating types be larger toward the wood than back out through the coating, leading to water condensing beneath the cladding [15,16]. This may be a problem especially if coupled with insufficient ventilation.

With the correct wood species, the natural degradation of a façade can be significantly slowed down but selecting a correct surface treatment is still the most important factor for securing the resistance of wood against weathering [17]. Wood in exterior applications can be protected by several means, but methods such as plasma treatment, chemical modification, or impregnation all have their downsides, such as high cost or lengthy preparation processes. Conventional solutions such as low-VOC coatings may have inadequate water-resistance, and even novel nanoparticle-added formulations may possess environmental risks [18]. Surface charring, however, is a very low-impact process that only requires an input of energy. A coating is composed of the actual coating, the boundary layer between coating and substrate, and the substrate itself, in this case, the wood [19]. A char surface can be considered similarly, even though the "coating", i.e., the char, is formed out of the substrate itself. Between the char and the wood lies a boundary layer, or the transition zone, the thickness of which depends on modification method. The adhesion between coating and wood is key, and fatigue and failure cause a coating to delaminate and peel. Decaying fungi preferentially colonize the layer between the coating and the wood, leading to further failure [20]. As with coatings, the transition zone in charred wood often seems to be the cause of problems, as it has been shown that cracks propagate from this zone [21], and a thicker zone may increase cracking in weathering [5].

Kymäläinen et al. [6,7] determined certain sorption properties for softwoods modified by contact charring with and without simultaneous compression. The contact angles were greatly improved and absorption in water floating reduced due to increased hydrophobicity. However, in some modification regimes the liquid water absorption experiments showed increased dimensional instability measured by cupping. The contact charring modification method presented in this study is based on these previous experiments, combining the best empirically determined processing parameters for optimal performance, without the compression that was suspected to lead to excessive cupping and cracking. Flame charring, on the other hand, is a process more difficult to control, as species- and sample-specific factors greatly affect the result. However, the larger specimen size required mimics real life situation, as entirely flawless wood samples cannot be used. Moreover, the modern producers use this method, further emphasizing the need for comparable information on its performance. The aim of this study is to compare two charring techniques: contact charring that creates a smooth, hard finish with a high contact angle and reduced sorption [7], and gas flame charring that creates a highly degraded and porous, but a chemically inert surface [5]. The goal is to further compare the characteristics of two softwoods (pine and spruce) and two hardwoods (birch and aspen) to gain information on the responses of different wood species to charring, and also to investigate the potential of woods currently less used for façades. As the char surface may theoretically be compared to a coating, a comparison is also made in practice. In addition, linseed oil is used to study the effect of a very simple, organic additive that could possibly be used to fix the friable carbon layer in the flame-charred samples, and further increase the hydrophobicity of the modified surfaces.

## 2. Materials and Methods

### 2.1. Materials and Sample Preparation

Mature softwoods Scots pine (*Pinus sylvestris* L.) and Norway spruce (*Picea abies* (L.) Karst), and hardwoods silver birch (*Betula pendula* Roth.) and trembling aspen (*Populus tremula* L.) were used as the research material. All the material was sourced from Southern Finland. Only sapwood boards were chosen with as straight grain as possible. The densities (unmodified, at 65% RH, 20 °C) were 0.51, 0.41, 0.62, and 0.55 g/cm$^3$, respectively. Contact charring was implemented on a stainless-steel hot plate set to 320 °C, with a weight of 16 kg, for a duration of 30 min. The samples were cut to 100 mm × 100 mm prior to charring, limited by hot plate dimensions. For flame charring, one-meter boards were used. Butane gas torch was used to create a consistent crack pattern. This usually took 2.5 to 3 min. Surface temperatures of 800–1100 °C were measured beneath the flame with thermocouples. Modification was always implemented on the pith side of the wood. Examples of surfaces are shown in Figure 1. Char depths, measured from several cut-outs with an Olympus SZX10 microscope (Olympus corp., Tokyo, Japan), were 1.1–1.3 mm for flame-charred specimens and 2.0–4.8 mm for contact-charred specimens. The value is a combination of char and transition layers and based on color differences. For flame-charred samples, the shift from char to unmodified wood is sharp and easily distinguished, but separation of the thermally modified transition layer between char and wood is not possible without use of high magnification devices such as SEM. On the contrary, separating char from the gradual transition zone on contact-charred wood is very difficult; thus, the combined thickness is given.

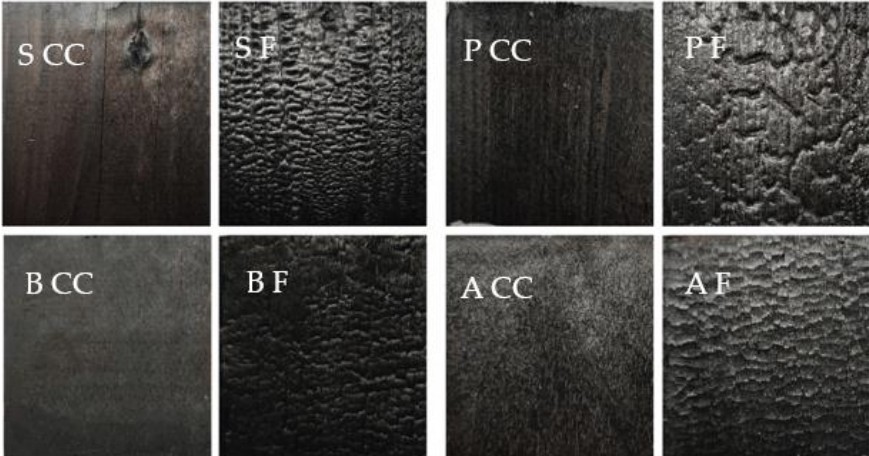

**Figure 1.** Contact-charred (CC) and flame-charred (F) spruce (S), pine (P), birch (B), and aspen (A).

The boards were cut to 100 mm × 50 mm (tangential × radial) for sorption experiments, excluding parts with uneven charring result (rounded edges, inadequate thickness, etc.,) or large knots. The contact-charred samples were cut in half for this purpose. Six replicates were prepared per modification (Table 1). Replicates also included linseed oil covered series, where two layers of oil were applied as instructed by the manufacturer (Woodcare.guide, Rakennuskemia Oy, Hyvinkää, Finland). Painted references were prepared using a water-soluble black acrylate exterior wood paint (Pika-Teho, Tikkurila OYJ, Vantaa, Finland). The samples were lightly sanded and painted once with an acrylate primer (Ultra Primer, Tikkurila OYJ, Vantaa, Finland), followed by two coats of paint as instructed by the manufacturer. The coating thickness was 0.05 mm, and about 0.1 mm with the primer. The samples were sealed from five surfaces using a double layer of flexible acetic acid-based silicone (Ardex SE, Ardex Gmbh, Witten, Germany). The initial MC was between 8 to 12%.

**Table 1.** Wood species, modifications, and abbreviations.

| Wood Species | Wood Species Code | Modification | Modification Code |
|---|---|---|---|
| Aspen | A | Unmodified/+ oiled | R/R-O |
| Birch | B | Painted | P |
| Pine | P | Contact-charred/+ oiled | CC/CC-O |
| Spruce | S | Flame-charred/+ oiled | F/F-O |

### 2.2. Water Vapor Uptake, Permeability and Moisture Exclusion Efficiency

The apparent vapor permeability for the wood surfaces was determined from the consecutive mass measurements which were performed to define the moisture content at a certain relative humidity (RH). The samples (six per series) were placed in a Rumed 5100 climate chamber (Rubarth Apparate GmbH, Laatzen, Germany) at a constant temperature of 20 °C. The samples were first conditioned at 30% RH, after which the RH was increased to 65% followed by 95%. The duration of a respected step was 14 days, after which the weight and the cup on pith side were recorded. After the experiment, the water uptake at each step (permeability) was calculated as $g/m^2$. As presented by Feist et al. [14], the moisture exclusion efficiency (MEE, %) of the surface was calculated as mass difference compared to unmodified reference,

$$\text{MEE} \ (\%) = \frac{U - C}{U} \times 100 \tag{1}$$

where U stands for mass of moisture adsorbed by uncoated (unmodified) reference and C for mass of moisture adsorbed by coated (modified) sample at same condition.

### 2.3. Liquid Water Uptake

The experimental setup followed the standard EN 927-5:2006 with the exception of sample size and lack of pre-conditioning (plus, as cupping was measured, the growth rings were not cut, i.e., the samples were cut across the grain). The samples sealed from 5 sides were preconditioned at 65% RH, 20 °C, after which they were set to float face down in a container with water. The mass and cupping were measured after 0, 6, 12, 24, 48, 72, and 96 h, after which the samples were oven-dried for MC determination. Moisture uptake was measured from increase in mass after each step and converted to $g/m^2$.

### 2.4. Dimensional Changes during Sorption

Cupping of the samples was measured to evaluate the effect of wetting on the dimensional stability of the modified surfaces. The cup was determined by a digital dial indicator (Mitutoyo IDU-25 Digimatic indicator, Mitutoyo Corp., Kawasaki, Japan) after the water vapor sorption experiment. The indicator was attached to a flat surface and values were recorded at the lengthwise center of the sample to an accuracy of 0.02 mm. To avoid water evaporation while moving the liquid water sorption samples from one laboratory to another, these samples were measured in situ with a similar handheld dial indicator.

### 2.5. Data Processing and Statistical Analysis

Statistical analysis was run to determine dependencies between modifications and measured responses. Equality of variances was tested beforehand for each group to ensure correct method, and normality of data with Kolmogorov–Smirnov/Shapiro–Wilkes tests. Despite having equal sample sizes, all investigated groups returned unequal variances and the data were not normally distributed. Therefore, Welch's analysis of variance (one-way) was used in combination with Tamhane T2 (results compared with Dunnett T3) post hoc analyses for pairwise differences and multiple comparisons, respectively. The permeability/floating values showed large variances, so Brown–Forsythe was also utilized for comparative purposes. IBM SPSS statistics version 28 was used to run the analyses. Because of a small sample size in the diffusion experiment ($n = 2$), a two-sample t-test was utilized.

## 3. Results

### 3.1. Permeability and Moisture Exclusion Efficiency

Figure 2 shows the water vapor uptake, i.e., permeability of the surface at 95% RH. The modifications affect all wood species similarly, with references showing the highest permeability at adsorption. Oiled references or contact-charred samples do not differ from unoiled ones (R, CC vs. R-O, CC-O). The modifications F vs. F-O significantly differ with hardwoods (plus spruce), as does the permeability between CC and F/F-O (Table S2). For softwoods, there seems to be no differences. Painting does not have a straightforward effect but somewhat reduces vapor uptake for birch and spruce in comparison to unmodified reference. The permeability drops noticeably for contact-charred samples, but further oiling has negligible effect within this regime. In comparison to reference, the measured values of CC and F modification differ significantly for all wood species, but for spruce there are no significant differences in R vs. F.

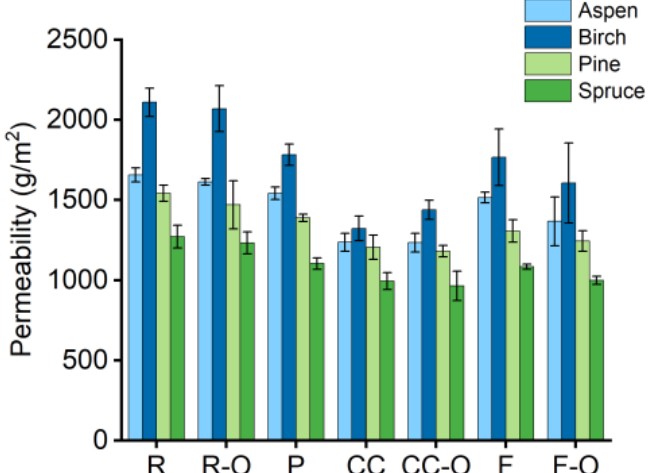

**Figure 2.** Water vapor permeability ($g/m^2$) of aspen, birch, pine and spruce with reference to mass at 30% RH. R = reference, P = paint, CC = contact-charred, F = flame-charred, O = oil added to respected surface. Error bars depict standard deviation.

The permeability depends significantly on species, within modification, in all modification regimes (at $p < 0.05$; Tables S1 and S2 in Supplementary Materials). The relative moisture permeability showed consistent decrease within all modifications within a respected modification, when compared to unmodified reference. For aspen, the permeability, in comparison to reference, was smallest for modifications ACC and ACC-O (about 25% decrease) and highest for AP and AF (about 8% decrease from reference). For birch, BCC also presented smallest permeability (37% reduction), and values of BP and BF were again of similar magnitude (about 15% decrease). For pine, PCC was the smallest with a 23% reduction to reference. For spruce, the permeability is highest for SP (13% less to reference) and smallest for SCC-O with 25% less to reference. Post hoc comparison revealed significant differences between modifications (within species) in regimes R vs. CC and CCO. Painted surface permeability differed significantly from CC on aspen, birch, and pine, but further oiling (CCO) did not affect aspen. Flame charring caused a change in permeability (R vs. F) on aspen, birch, and pine, whereas F vs. FO was significant only on spruce. The differences between modifications were slightly less significant on spruce than on other studies wood species (detailed comparisons in Supplementary Materials, Tables S3 and S7).

Moisture exclusion efficiency was determined from the mass gained in adsorption (30 to 95% RH) and the results are presented in relation to the unmodified reference sample of the respected series (Figure 3). A higher ratio therefore represents a higher MEE, i.e., a stronger barrier effect against water vapor. The contact-charred samples have a relatively high MEE, birch showing very high values. Painted surfaces give very average MEE values, but the standard deviation between samples is low and consistent. The

charring modifications show much higher between-sample variation, especially in the flame charring regimes. For practical purposes the results suggest that significant barrier effects can be reached with all charring modifications (Table S1). Birch stands out as it did in permeability measurements, but only within the contact charring modification. Post hoc results (Tables S4 and S8) again reveal similarities between aspen and pine within the contact charring modification. Species affects the measured MEE significantly in modifications P, CC, CC-O, and F, although the difference in F is significant only between aspen and spruce. Hardwoods present overall slightly more statistically significant results in comparison to softwoods when comparing the modifications to each other within the respected species.

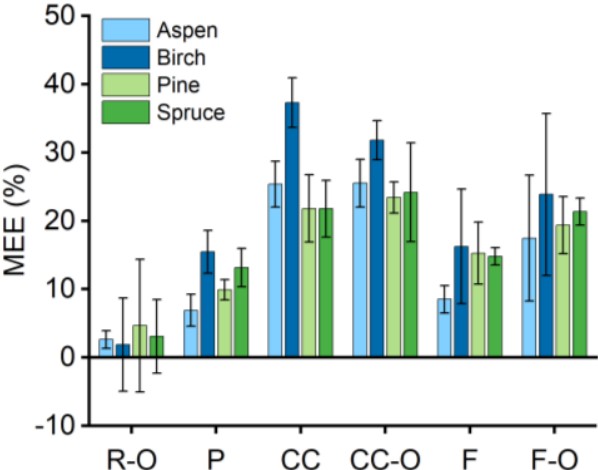

**Figure 3.** Moisture exclusion efficiency (MEE) calculated from mass gain at adsorption between 30 and 95% RH, in relation to unmodified reference of respected species. Error bars for SD.

### 3.2. Liquid Water Uptake

The liquid water uptake was measured with freely floating samples for a total period of 96 h (Figure 4). As with water vapor adsorption, liquid water sorption was significantly connected to modification, but between species the results were significant only for F and F-O. The post hoc analyses show there is a significant effect when comparing unmodified reference to other modifications, but less so when comparing modifications to one another (Table S5). There is little effect for wood species, but the consistent liquid water uptake, with both F and F-O showing great reduction on spruce is significant. The reduction in water uptake was comparable to simple addition of oil on reference surface, with a reduction of up to 79%. Oiling decreased the uptake of spruce in all cases (compared to respected unoiled modification). The magnitude varied from an improvement of about 9% (SF-O vs. SF) to about 84% (SCC-O vs. SCC). Other wood species presented values between these. Especially pine, aspen, and birch seemed to benefit from additional oil coating. The SCC-O modification was the most successful in terms of absolute uptake, with 93% reduction to SR. Unfortunately, the CC-O values could not be recorded for other wood species, as the combination of contact-charred surface and oil caused the sealant to peel, letting water penetrate the other sides in addition to the modified surface. Similarly, some flame-charred and oiled samples were discarded due to peeling and replaced with intact specimens. In addition, the painted referenced showed heavy cracking of the silicone applied to the sides presumably because of the water retention of the painted layers causing sample swelling. This happened at 48 h for birch, 48 to 72 h for spruce, and 72 h for pine, making especially the birch results somewhat unreliable. Painted aspen experienced no such problem when inspected visually, although the measured increase in mass was very high. Silicone was chosen as the sealant because of easy application and flexibility. The formulation also includes a fungicide, which was considered important in the climate chamber experiment, where the samples were exposed to conditions favorable to fungal growth for extended

periods of time. However, the moisture exclusion efficiency was not as good as for epoxies and coatings used in standardized experiments (although the reference values of a silicone surface measured prior to experiments were within acceptable limits). Thus, the absolute values seem very high in comparison to some of the values reported in the literature. The values are however well comparable to each other and reveal the differences between modifications reliably (except for painted birch due to cracking of sealant).

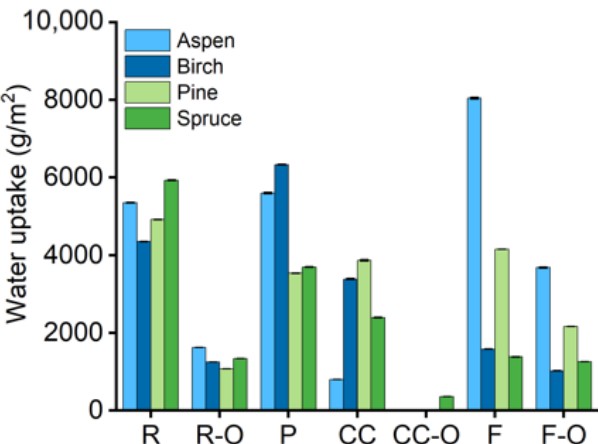

**Figure 4.** Water uptake ($g/m^2$) of aspen, birch, pine and spruce in liquid water floating at 96 h. Error bars for SD.

### 3.3. Dimensional Changes in Sorption

The magnitude of dimensional changes was evaluated from samples exposed to both water vapor and liquid water in the floating experiment. The results for cup in mm, measured from modified face of samples after exposure to 95% RH are shown in Figure 5. The effect of contact charring is obvious, as it increases cupping in relation to all other modifications. Statistically, AR, AP, and AF are very similar to corresponding pine samples. The initial cup measured after conditioning at 30% RH did not reveal significant differences between modifications on softwoods, unlike on hardwoods. It was therefore seen that the modification affects the cupping of hardwoods at lower humidity, although the magnitude was not very large in absolute terms. Cupping of spruce and pine, on the other hand, is significantly different from the rest at the high RH. At 95% RH the cupping is highest for all hard- and softwood contact-charred samples.

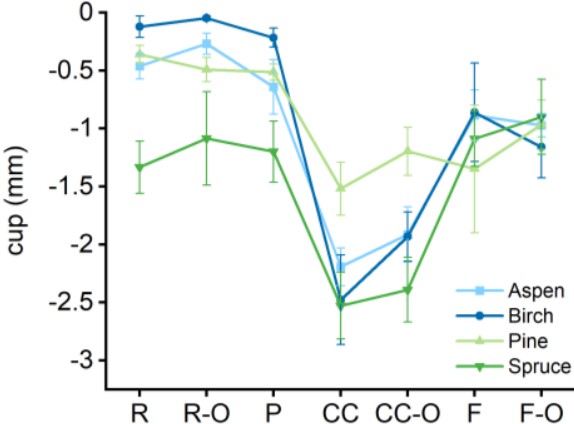

**Figure 5.** Cupping of aspen, birch, pine, and spruce due to water vapor sorption measured from modified face after conditioning at 95% RH, 20 °C.

The cupping of samples exposed to liquid water sorption was measured in situ to avoid evaporation of water while moving the samples. The measurement practice in

general contains error sources since the technique relies on a single point that is pressed against the surface. The exact measuring point, and structural differences, such as grooves and crevices on the char, impact the results. The measurement table used for the vapor sorption specimens minimized these errors, but the handheld indicator used for liquid water absorption specimens proved more problematic because of issues with both measuring location on the specimen, applied pressure during measuring, as well as device calibration. Therefore, to simplify the results and minimize error sources, the values were categorized according to the magnitude of measured cup. The classes ranged from −3 to +3 in accordance to measured cup that ranged from more than −3 mm and more than + 3 mm. The frequency was calculated according to these seven classes. At 0 h, most of the samples were situated in classes 0 and +1 (−0.99 mm to +1.99 mm) (Figure S1 in Supplementary Materials). Floating for 96 h increased the number of samples falling to other categories. Most prominent outliers were found within the charred samples. Four out of six PCC samples showed a negative cup over −4 mm at 96 h. Half of SCC also showed strong negative cup, as did several variants of AF. The negative cup was measured from SCC and PCC already at 0 h, as it is a modification effect, but moisture exposure greatly increased the magnitude. The hardwoods birch and aspen behaved in a different manner, where the cup was partly to other direction, i.e., towards the unmodified back of the samples (positive cup, "bow"), and the magnitude was lower. Most of these samples were found in classes 0 and +1 also after the experiment, translating to a rather moderate cup.

## 4. Discussion

Water uptake has important consequences for both wood and coatings used for wood. Dimensional movement leads to wood distortion, splitting and cracking, and promotes decay. Additionally, a high MC exerts mechanical stresses and induces molding and adhesion impairment on coatings [22,23]. The moisture movement in wood below the fiber saturation point is governed by at least two different mechanisms: the water-vapor diffusion in the cell lumen and pit openings and bound-water diffusion in the cell wall substance [24]. However, bound water transport is of little significance in large wood specimens due to slowness of the process [25,26]. The vapor diffusion through the barrier (coating/charred surface) is thus the main process to consider. Gravimetric methods are effective in evaluating the transport of water molecules through the coating into the wood. The cup test is a standardized means to measure the mass transfer and permeability, as well as the diffusion coefficient through a coating. However, the technique is very sensitive to unevenness of coating, as well as the presence of a transition layer [27], both of which are found in a charred wood specimen. In the case of char, the properties of the underlying wood also have a great effect on the results. With this in mind, the measurement of mass increase and decrease in varying humidity in the climate chamber was seen as reliable method to analyze the permeability of the surface. In wood char the moisture movement is similar to wood, but certain structural, chemical, and physical aspects differ, causing a decreased affinity to water. Water vapor diffusion in wood char is greatly hindered because of, e.g., removal and conversion of carbohydrates [28], increased hydrophobicity and changes in surface functional groups [29–31], increasing rigidity of the material that restricts swelling, changes in water molecule clustering behavior and specific surface area [32]. Moreover, the ability of water molecules to attach to cell walls decreases as a function of modification severity. Consequently, the lower temperature used in contact charring results in a smaller degree of structural degradation in comparison to flame charring, albeit the ingress of water molecules is significantly slower than on unmodified references. First, the wood composition has gone through extensive changes similar to, but slightly further than, thermal modification (temperature range 150–240 °C [33], which already effectively reduces accessibility of the cell walls [34–36]. Second, the surface plasticizes in the process and becomes hard and hydrophobic [6,7,9]. This is partly because of extractives that affect the sorption. In contact charring the extractives are vaporized, but

due to tight contact to the heated plate, they are likely to condense on the surface of the wood at least partly, which further promotes hydrophobicity.

The permeability showed a consistent trend among the studied wood species, with the overall lowest permeability values measured from CC. The unmodified references (R) were comparable to painted surfaces on all species and to flame-charred surface on aspen, pine, and spruce. This is not surprising, as waterborne coatings including the modern acrylic dispersion paints are generally very permeable [23]. Their film-forming characteristics ensure blocking of pores, and rather deep penetration has been reported on pine [37]. On impermeable species such as spruce and meranti the effect was less pronounced, and moisture uptake was similar to uncoated references [37,38]. Although we opted to use a common acrylic primer, it is noteworthy that acrylic coating is often recommended to be applied on top of an alkyd primer for satisfactory protection against water [39]. Linseed (flax) oil, on the other hand, is the most used natural oil for wood applications in Europe [40]. Linseed oil can be used on its own, or as an ingredient in alkyd or polyurethane paint systems [41]. The oil penetrates wood pores and increases hydrophobicity. There is some indication of the effect of substrate's chemical composition, most importantly the extractives content. In Arminger et al. [40], film formation on oak wood was found inhibited, whereas complete polymerization was seen when applied on beech wood. It is possible that the overall plasticization and hardening, but also the condensation of extractives on a contact-charred surface inhibited penetration and/or polymerization of oil, which then led to peeling of the sealant in CC-O samples in water floating. In this paper, the addition of linseed oil in the study regime was motivated more by the desire to fix the friable carbon layer of the flame-charred surfaces than to increase hydrophobicity. Indication of this kind of binding of char was seen in natural weathering experiments by Kymäläinen et al. [5]. Here, oiling increased the variation in MEE values of flame-charred and oiled surfaces. The char is highly porous, and it is likely that the final distribution of oil within the porous matrix is not even, which promotes high standard deviation values. Dipping would perhaps be better than application by brush, but the consumption of oil would also increase manifold. The permeability values of AF-O and BF-O also show this higher variation. The exact nature of the flame-charred surface depends on the wood species. Aspen especially presented a rather small-scale flaking, whereas the spruce and pine samples had surfaces with larger rectangular crack patterns (see Figure 1). The birch surface was visually rather similar to aspen, but as a denser species it produces a more compact char. This brings about an improvement in liquid water sorption in contrast to aspen. However, aspen seemed to benefit quite much from the oiling, which is likely translated as the overall decrease in hydrophobicity. Birch and aspen are both diffuse-porous hardwoods with a few ultrastructural differences mostly related to the structure of perforation plates within the vessels (scalariform on birch, simple on aspen). The higher density of birch is a probable stabilizer in comparison to aspen. Pine showed similar behavior to aspen in several cases, especially in terms of oiling. This is likely to be related to the open structure of pine as well as aspen. The large cross-field fenestriform pits increase moisture transport within pine wood and likely benefit from the barrier effect brought about by linseed oil.

In terms of absolute values of MEE, the flame charring modification is comparable to a double layer of acrylic paint on primer. The CC modification performed even better. Same trend is observable in the permeability results, as well as liquid water sorption. This stems from the overall reduced adsorption capacity of low-temperature charred wood [7,32]. The high permeability of acrylic dispersion paints also has a direct effect [14]. Char has a greater porosity to unmodified wood because of shrinkage of fibers and removal of components due to thermal degradation that result in formation of a more open structure [28]. A higher water absorption in liquid water exposure would thus be expected, but a decrease in water uptake was recorded for all charring modifications (except flame-charred, unoiled aspen). Gosselink et al. [28] argued, that low-temperature carbonized (at 275 °C) water-immersed wood reaches almost the same moisture content as untreated wood because of its structure.

However, wood that has been modified below 300 °C is not "true" char. Thus, water uptake of chars created at 370–620 °C was characterized as a combination of both porosity and overall hydrophobicity, when these two factors were separately investigated [42]. Very hydrophobic wood chars were created also at a relatively low temperature of 400 °C and an increase in the water retention was seen with increasing treatment temperature up to 800 °C, indicating the formation of a sufficient microporosity [43]. Similar conclusions were drawn by Kymäläinen et al. [32] regarding torrefied wood. The authors measured a consistent decrease in the specific surface area of torrefied (=charred) wood samples below 300 °C, but a breakdown of the model took place for samples modified at 450 °C. This led to the speculation on pore blocking by degradation products or a formed microporosity too small for water penetration. Pulido-Novicio et al. [44] reported that no microporosity had been developed for sugi carbonized at 400 °C, but high tar formation led to blocked porosity. Blockage caused by other degradation products was also seen. Tar formation increases substantially when the secondary degradation pathway of cellulose takes place above 300 °C [45]. It is therefore likely that the reduced water absorption in samples described in this paper is a result of increased hydrophobicity, rigidity, porosity changes as well as some degree of pore blocking. For a contact-charred sample, the very high contact angles [6,7] also reduce the uptake of the surface by creating negative capillary forces.

The coating permeability has great implications on dimensional changes in the wood substrate, both swelling and shrinkage in absorption and desorption [23]. The moisture content within a charred wood piece is asymmetric because of differing vapor permeabilities and barrier effects of the surface and the interior. Due to this, differential deformations also take place, and eventually lead to stress-induced cracks [46]. Cupping is an important characteristic of wooden cladding boards, since it causes loosening of attachments and cracking of the boards [47], and a cracked surface will also deteriorate faster than an intact one [48]. A flexible coating can restrict/control small scale checking, but the coating will rupture when too high tensile and compressive stresses are applied. A char layer, on the contrary, is brittle by nature, but the hypothesis is that overall reduced moisture uptake and reduced swelling also reduce cupping distortion. Cupping on the pith side (negative cup) is a characteristic phenomenon in drying, and it is exaggerated in contact charring because of a steep temperature and, consequently, moisture gradient. The surface is left in set compression due to dehydration and shrinkage. Usually, when wood is wetted, the surface swells more in comparison to the core, which is still in dry state [49]. In a tangentially cut piece of wood, this will reverse the stresses and the wood will bow, creating tensile stresses on the pith side surface. What is interesting in the contact-charred samples is that they continue to cup in "wrong direction" during wetting, whereas for references and the painted samples the direction is corrected (levelling out towards zero and above into a positive cup, "bow") as the annual rings seek to straighten. The positive cup was not seen in tabletop measurements after water vapor sorption experiment, but this is more a property of the measuring table. The dial is located beneath the table, with a zero point on table level, when no exception is made between zero cup and positive cup when the measuring point is at table level. Therefore, measurements from the back side should also be reported to give a more detailed account of the "bow" effect. Regardless, the contact charring modification stood out in terms of cup in both sorption experiments. In liquid water sorption, the cup was greatest with softwoods SCC and PCC, whereas the values for hardwoods were more scattered. In water vapor sorption, the pine samples showed least distortion. Also, it was clear that in comparison to CC/CC-O, the F/F-O modification was better in terms of form stability. However, CC-O showed less cup in comparison to CC, although this was significant only for hardwoods. This is likely to be a result of the improved hydrophobicity in comparison to unoiled samples. Overall, the hardwoods were more dimensionally stable than the softwoods, also in the CC modification. Finally, statistically the results indicate that modification has an effect within species, and in most cases also species has an effect within a modification. This was pronounced for permeability and cupping results. The sample size was rather small, which causes some uncertainties, but it has been shown that

the Welch's ANOVA is a rather robust method. It has more power within the chosen level of alpha, although it has been recommended to use the Brown–Forsythe test in case where the data are skewed. This was indeed the case here, but the returned *p*-values did not affect the significance at the chosen level of 0.05 regardless of method.

## 5. Conclusions

The results show that contact charring brings about clear benefits for wood in terms of sorption properties—water vapor as well as liquid water uptake are reduced, and the moisture exclusion efficiency improved. The measured values within the flame charring modification are comparable to an acrylic coating in terms of liquid water sorption, permeability and MEE, whereas contact charring further improves these characteristics. The results for flame charring are more dependent on wood species, as e.g., aspen showed a rather high liquid water uptake in contrast to other species and modifications, although applying oil improved the results. In several cases the flame charring was equal to double layer of acrylic paint on primer, or to unmodified reference. Spruce showed an overall good performance regarding sorption of contact and flame-charred surfaces, and it may be of interest to investigate also other impermeable species. Coating the charred wood surface with a natural oil-based additive shows promise, as it increases the hydrophobicity, but also renders the surface easier to handle (especially F-O). The dimensional stability of flame-charred woods was better than that of the contact-charred samples. However, there was indication of reduced cup when oil was added to the contact-charred (hardwood) surfaces. The hardwoods showed overall less distortion than softwoods, but the effect of oil is still an interesting subject to investigate further. As a conclusion it is rather safe to say, in the context of the measured properties, that charring modifications improve the sorption properties of studied wood species to a level of common paint solution or above, although the dimensional instability of the contact-charred samples is an issue that needs further research.

**Supplementary Materials:** The following supporting information can be downloaded at: https://www.mdpi.com/article/10.3390/coatings12091273/s1. Table S1: Welch's ANOVA for equality of means on permeability, moisture exclusion efficiency (MEE), liquid water sorption and cupping after water vapor sorption, within species between modifications, Table S2: Welch's ANOVA for equality of means on permeability, moisture exclusion efficiency (MEE), liquid water sorption and cupping after water vapor sorption, within modifications, between species, Table S3: Water vapor permeability at 95%RH. Within species comparison of modification effect, compared modifications marked with blue color differ significantly by Tamhane's T2 at 5% significance, Table S4: MEE (%) Within species comparison of modification effect, compared modifications marked with yellow color differ significantly by Tamhane's T2 at 5% significance, Table S5: Liquid water uptake at 96 h. Within species comparison of modification effect, compared modifications marked with green color differ significantly by Tamhane's T2 at 5% significance, Table S6: Cupping at 30% and 95% RH. Within species comparison of modification effect, compared modifications marked with orange color differ significantly by Tamhane's T2 at 5% significance. Darker color stands for significant differences at 95% RH, whereas lighter color for differences seen at 30% RH, Table S7: Water vapor permeability at 95% RH. Comparison between species within same modification, variables marked with blue color differ significantly by Tamhane's T2 at 5% significance, Table S8: MEE (%) Comparison between species within same modification, variables marked with yellow color differ significantly by Tamhane's T2 at 5% significance, Table S9: Liquid water uptake at 96 h. variables marked with yellow color differ significantly by Tamhane's T2 at 5% significance, Table S10: Cupping at 30% and 95% RH. Comparison between species within same modification, variables marked with yellow color differ significantly by Tamhane's T2 at 5% significance, Figure S1. (a) Frequency of samples classified by magnitude of cup: –3 (equal or more than –3 mm); –2 (–2 to –3 mm); –1 (–1 to –2 mm); 0 (–0.99 to 0.99); 1 (1 to 2 mm); 2 (2 to 3 mm); 3 (equal or more than 3 mm) measured at 96 h into liquid water floating (from modified surface). (b) schematic illustration of cupping when measured from pith side/modified face of the sample. Positive = "bow", negative = "cup".

**Author Contributions:** Conceptualization, M.K. and J.D.; methodology, M.K. and J.D.; formal analysis, M.K. and J.D.; writing—original draft preparation, M.K.; writing—review and editing, M.K., J.D. and L.R.; visualization, M.K.; supervision, L.R. All authors have read and agreed to the published version of the manuscript.

**Funding:** M.K. was funded by Academy of Finland Postdoctoral grant for project CHARFACE.

**Institutional Review Board Statement:** Not applicable.

**Informed Consent Statement:** Not applicable.

**Data Availability Statement:** The data used in this study is available on request.

**Acknowledgments:** J.D. is grateful for the European Union's Horizon 2020 research and innovation programme under Grant Agreement No. 952314.

**Conflicts of Interest:** The authors declare no conflict of interest.

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
