# Peer review of "Moisture Sorption of Wood Surfaces Modified by One-Sided Carbonization as an Alternative to Traditional Façade Coatings"

_coatings, doi:10.3390/coatings12091273_

Round 1

Reviewer 1 Report

For coatings-1807327, Comments:

1. Dimension stability of wood surfaces by one-sided carbonization after long-time water soaking is very important. I suggest adding this part of the data to the manuscript.

2. How to control the carbonation depth of wood with different densities. Missing data on wood carbonation depth in this manuscript.

3.  What are the mechanical properties of the wood surface after carbonization. Will the mechanical properties of the wood itself be affected?

4. Surface carbonization may also enhance other wood properties such as antibacterial, mildew resistance, insect resistance, etc. It is suggested that the author can explore in the future and does not need to be displayed in this article.

5. Line 144, should be MEE(%)

Line 302, should be 96 h

Line 339, should be 20 oC

Author Response

Reviewer #1

  1. Dimension stability of wood surfaces by one-sided carbonization after long-time water soaking is very important. I suggest adding this part of the data to the manuscript.

- The measurement procedure and data are explained on line 341-362. The figure describing the results was added in Supplementary materials, Figure S1.

  1. How to control the carbonation depth of wood with different densities. Missing data on wood carbonation depth in this manuscript.

- Thank you for pointing this out, information in char layer thickness was added (Lines 114-123). Denser wood tends to carbonize deeper, which is discussed later in the manuscript.

  1. What are the mechanical properties of the wood surface after carbonization. Will the mechanical properties of the wood itself be affected?

- The carbonization makes the surface soft (flame charring) or hard (contact charring). This is mentioned in several parts of the discussion, but not directly measured as the focus of this paper is on sorption properties. Naturally, the surface mechanical properties affect the sorption also, and this is dealt with in the discussion with comparison of flame and contact charred surfaces. The physical and mechanical changes induced by carbonization are evaluated more in depth in an upcoming manuscript still in preparation.

  1. Surface carbonization may also enhance other wood properties such as antibacterial, mildew resistance, insect resistance, etc. It is suggested that the author can explore in the future and does not need to be displayed in this article.

- This is true, and some of these properties are a direct consequence of pyrolysis, while others depend on the moisture interactions of the substrate and, to our opinion, deserve to be mentioned in discussion. A manuscript dealing with fungal growth on carbonized wood surfaces is being prepared at the moment.

  1. Line 144, should be MEE(%)

- Corrected

Line 302, should be 96 h

  • Corrected

Line 339, should be 20 oC

  • Corrected

Reviewer 2 Report

Dear Authors, many thanks for this interesting manuscript. I am with the main points of this manuscript, but I have some suggestions for potential improvement of this article.

I understood the different modification setups: flame charring and contact charring. The later with hot plate (320 Degrees) but what was the temperature with first method. I guess this method is described in an other article in detail, please refer to this sources.

The methods and investigations are described very well. Data statistical analysis is described and ok. But why, the diagrams shows something else than described in this chapter of statistical analysis (the diagrams shows error-bars, but this is described nowhere).

I am not happy with Table 2 and 3. Yes you did a statistical analysis. But neither in the text, nor in the diagrams this was used. I think this two tables should be placed into the annex or in this paper this information must be used better in diagrams and text.

The results f diffusion coefficient are impossible to use. I guess there were some problems in running the experiment or it is impossible to use this method for such small samples cuted from the samples. This results need to be repeated and can’t be published in this way (an indicator is, that this results are not included in the conclusion chapter). Please skip this from this manuscript, repeat this, prove this results and publish this in a separate short note.

Author Response

Reviewer #2

Dear Authors, many thanks for this interesting manuscript. I am with the main points of this manuscript, but I have some suggestions for potential improvement of this article. I understood the different modification setups: flame charring and contact charring. The later with hot plate (320 Degrees) but what was the temperature with first method. I guess this method is described in an other article in detail, please refer to this sources.

  • The measured temperature range was added (lines 112-113).

The methods and investigations are described very well. Data statistical analysis is described and ok. But why, the diagrams shows something else than described in this chapter of statistical analysis (the diagrams shows error-bars, but this is described nowhere).

  • Thank you for this comment. We aimed to show the relations between modifications and species in a clear manner in the diagrams, with the tables supporting the results and discussion on the significance of the findings.
  • Mention of error bars depicting standard deviation was added to the diagrams.

I am not happy with Table 2 and 3. Yes you did a statistical analysis. But neither in the text, nor in the diagrams this was used. I think this two tables should be placed into the annex or in this paper this information must be used better in diagrams and text.

  • The diagrams were drawn only with standard deviation because of clarity. If significant differences were added, the pictures would become much more difficult to read. Therefore, we added the two tables showing comparisons between species (within modification) and between modifications (within species) for a simple overview of significant differences. The Welch’s ANOVA is a one-way comparison and therefore two different test setups were needed, thus two tables. The tables are referred to in several places of the results section – without the information it would be rather questionable to state a finding to be significant. However, we agree that the tables are more suitable to be presented in Supplementary materials and can now be found as Tables S1 and S2.

The results of diffusion coefficient are impossible to use. I guess there were some problems in running the experiment or it is impossible to use this method for such small samples cuted from the samples. This results need to be repeated and can’t be published in this way (an indicator is, that this results are not included in the conclusion chapter). Please skip this from this manuscript, repeat this, prove this results and publish this in a separate short note.

  • DVS can also be successfully used to measure diffusion of very small films. The Payne cell is used for this purpose, and we also attempted to use it, however, the char was too fragile to be placed on the cell. This is why we used the setup described in the paper. But, we do agree that too few measurements were made and will leave the results out as suggested, focusing on procuring more repeatable and reliable results in the future.

Round 2

Reviewer 1 Report

Accept in present form

Reviewer 2 Report

Thanks to the author team for improving the manuscript. I am fine with the suggested changes.